# Serum NMR-Based Metabolomics Profiling Identifies Lipoprotein Subfraction Variables and Amino Acid Reshuffling in Myeloma Development and Progression

**DOI:** 10.3390/ijms241512275

**Published:** 2023-07-31

**Authors:** Shona Pedersen, Morten Faarbæk Mikkelstrup, Søren Risom Kristensen, Najeha Rizwana Anwardeen, Mohamed A. Elrayess, Trygve Andreassen

**Affiliations:** 1College of Medicine, QU Health, Qatar University, Doha 2713, Qatar; 2Department of Health Science and Technology, Aalborg University, DK-9220 Aalborg, Denmark; mfmi16@student.aau.dk; 3Department of Clinical Biochemistry, Aalborg University Hospital, DK-9000 Aalborg, Denmark; srk@rn.dk; 4Department of Clinical Medicine, Aalborg University, DK-9000 Aalborg, Denmark; 5Biomedical Research Center (BRC), Qatar University, Doha 2713, Qatar; n.anwardeen@qu.edu.qa (N.R.A.); m.elrayess@qu.edu.qa (M.A.E.); 6Department of Circulation and Medical Imaging, Norwegian University of Science and Technology, NO-7491 Trondheim, Norway; trygve.andreassen@ntnu.no; 7St. Olavs Hospital HF, NO-7006 Trondheim, Norway

**Keywords:** multiple myeloma, monoclonal gammopathy of undetermined significance, serum diagnostic metabolites, Nuclear Magnetic Resonance, multivariate analysis, univariate analysis, pathway analysis

## Abstract

Multiple myeloma (MM) is an incurable hematological cancer. It is preceded by monoclonal gammopathy of uncertain significance (MGUS)—an asymptomatic phase. It has been demonstrated that early detection increases the 5-year survival rate. However, blood-based biomarkers that enable early disease detection are lacking. Metabolomic and lipoprotein subfraction variable profiling is gaining traction to expand our understanding of disease states and, more specifically, for identifying diagnostic markers in patients with hematological cancers. This study aims to enhance our understanding of multiple myeloma (MM) and identify candidate metabolites, allowing for a more effective preventative treatment. Serum was collected from 25 healthy controls, 20 patients with MGUS, and 30 patients with MM. ^1^H-NMR (Nuclear Magnetic Resonance) spectroscopy was utilized to evaluate serum samples. The metabolite concentrations were examined using multivariate, univariate, and pathway analysis. Metabolic profiles of the MGUS patients revealed lower levels of alanine, lysine, leucine but higher levels of formic acid when compared to controls. However, metabolic profiling of MM patients, compared to controls, exhibited decreased levels of total Apolipoprotein-A1, HDL-4 Apolipoprotein-A1, HDL-4 Apolipoprotein-A2, HDL Free Cholesterol, HDL-3 Cholesterol and HDL-4 Cholesterol. Lastly, metabolic comparison between MGUS to MM patients primarily indicated alterations in lipoproteins levels: Total Cholesterol, HDL Cholesterol, HDL Free Cholesterol, Total Apolipoprotein-A1, HDL Apolipoprotein-A1, HDL-4 Apolipoprotein-A1 and HDL-4 Phospholipids. This study provides novel insights into the serum metabolic and lipoprotein subfraction changes in patients as they progress from a healthy state to MGUS to MM, which may allow for earlier clinical detection and treatment.

## 1. Introduction

Multiple Myeloma (MM) is the second most prevalent hematological cancer—an incurable disease with diagnostic delays and multiple relapses [1,2,3,4,5]. It accounts for 1% of neoplastic diseases in high-income countries [3], with a global mortality of 106,000 cases yearly [6]. In MM, malignant plasma cell clones produce excessive amounts of specific immunoglobulin (M-protein) and light chains [7]. Multiple myeloma begins asymptomatically as monoclonal gammopathy of undetermined significance (MGUS) and progresses to bone pain, anemia, kidney dysfunction, and infections [8,9]. Surprisingly, the 5-year survival rate for people diagnosed at an early stage is over 77 percent [1]. This is partly due to heterogeneous chromosomal aberrations and a variety of mutations in a number of genes, making it extremely challenging to target the disease therapeutically [10]. Consequently, searching for early diagnostic biomarkers and innovative therapeutic targets is crucial for preventing multiple myeloma.

Furthermore, the molecular mechanisms sustaining the progression of the disease from MGUS to MM are poorly understood. MGUS and MM share an astonishingly similar genomic architecture [11]. Therefore, elucidating the metabolomic shift from asymptomatic to symptomatic MM may serve as a platform for mapping the dysregulated phenotype associated with this malignancy [12]. Recently, metabolomics, a quantitative measurement of all low-molecular-weight metabolites, is gaining momentum for diagnosing, classifying, making treatment decisions, and assessing treatment efficacy in cancer pathology and other disorders [8]. In addition, mounting evidence demonstrates that metabolomics profiling is well-suited for identifying prognostic and diagnostic markers in patients with hematological malignancies [13,14,15].

Nuclear magnetic resonance spectroscopy (NMR) and mass spectrometry (MS) have emerged as the two most common techniques in metabolomics research, each with advantages and limitations [16]. Over the last 15 years, the number of NMR-based metabolomics studies for mapping cancer development, progression, and treatment has risen [16]. This attribute is because NMR is a non-destructive, unbiased, quantitative method that requires minimal sample preparation and standardized and automated data processing [17]. In addition, NMR-based metabolomics platforms have several distinct advantages over MS-based platforms. Firstly, NMR is highly sensitive to the chemical environment and can provide information on molecules in a physiological setting [18]. Secondly, although NMR is less sensitive than MS, it is more applicable in clinical research because it is better suited for large-scale metabolomic studies [16,17,19].

Currently, our understanding of the altered metabolome in MGUS and MM is limited, with only a few publications describing the metabolic changes in MM. The first metabolic study on MM cells revealed that progression to MM depends on glutamine and glucose metabolism [20]. Additionally, it has been reported that an altered bone marrow metabolism is an early trait of MGUS development and is unrelated to the disease’s progression to MM [21]. Researchers have also suggested that peripheral serum and plasma can be applied to explore the metabolic phenotype between MGUS and MM [12,22,23,24]. Moreover, Steiner and colleagues demonstrated through targeted MS-based metabolomics that eight plasma metabolites differ significantly between MM and MGUS [8]. Researchers have shown significantly altered serum and plasma metabolites in MM or MGUS compared to healthy controls [12,22,23,24,25,26,27]. However, these studies focused primarily on the metabolic changes due to treatment response or at a single point in the progression of the disease. However, Ludwig and colleagues were the first to compare the plasma metabolic changes between controls, MGUS, and MM using ^1^H-NMR spectroscopy profiling. Still, they were unable to distinguish the premalignant from the malignant disease states in MM [21]. Interestingly, lipoproteins are becoming increasingly relevant as a prognostic factor in cancers, particularly in myeloma [28,29,30,31,32].

In this study, we investigate the global aberrations of metabolites and lipoprotein subfractions, specifically focusing on the comparisons between healthy controls, MGUS and MM. Our approach involved utilizing NMR-based metabolomics, employing 1H-NMR spectroscopy as the primary analytical technique. This research aims to refine our molecular understanding of this incurable disease and to identify candidate metabolites prior to clinical manifestations, facilitating a more effective preventative treatment.

## 2. Results

### 2.1. Clinical Characteristics of Study Populations

A total of 20 MGUS and 30 MM patients fulfilled the inclusion criteria for this study. Staging in accordance with the criteria for the International Staging System for MM and various other characteristics are shown in Table 1. The clinical data and biochemical parameters were previously presented in a study by Nielsen et al. [33]. Briefly, MM patients showed biochemical anomalies such as an increased serum protein, creatinine, C-reactive protein (CRP), M-protein, and decreased albumin, fibrinogen, and hemoglobin (Table 1 and Appendix A-Clinical data). In addition, several MM patients exhibited severe bone changes and elevated levels of plasma cells in the bone marrow.

### 2.2. Healthy Control vs. MGUS: Progression of MGUS Associated with Imbalanced Amino Acid Metabolism

To identify potential biomarkers linked to premalignant and malignant MM, 41 metabolites and 114 lipoprotein subfraction variables (Appendix A-NMR data) were analyzed using partial least squares discriminant analysis (PLS–DA). Figure 1A shows a significant differentiation between healthy subjects and MGUS patients, with a mean cross-validation error rate of 0.16. The PLS–DA analysis revealed 16 variables, mainly amino acids, with VIP scores > 1.0 that significantly differed between MGUS patients and controls (Figure 1B, Appendix A). ROC curves and boxplots for comparisons of the four metabolites with the highest AUC scores are displayed in Figure 1C. Lysine, formic acid, and leucine exhibited a remarkable AUC performance of 0.86 (95% CI = 0.75–0.97), AUC of 0.84 (95% CI = 0.71–0.97), and an AUC of 0.82 (95% CI = 0.69–0.94), respectively. The combined ROC analysis on the four perturbed markers revealed an AUC of 0.99 (95% CI = 0.95–1.00) (Appendix A). Only formic acid showed a positive fold change (FC) of 1.8, indicating elevated levels in the MGUS group. Alanine and isoleucine had AUC scores below 0.80 (0.77 and 0.78, respectively) and reduced levels in the MGUS group. Multiple pathways were altered in MGUS, as shown in Figure 1D. These pathways included phosphatidylinositol phosphate, glycerophospholipid and galactose metabolism, folate metabolism, prostaglandin formation, methionine, and cysteine metabolism, valine, leucine and isoleucine degradation, glycine, serine, alanine metabolism, urea cycle, and metabolism of arginine, glutamate, aspartate and arginine, lysine, lipoate, and biotin. Furthermore, we noticed a notable link between creatinine and alanine and between serum protein and formic acid. In our study, these metabolites have been recognized as indicators connected to the progression from a normal state to the early stage of monoclonal gammopathy of undetermined significance (MGUS). They may potentially contribute to developing multiple myeloma (MM) pathology (Appendix A).

### 2.3. Healthy Control vs. MM: Low Levels of Apolipoprotein and Cholesterol Are Prevalent in MM Patients

The PLS–DA plot depicted in Figure 2A distinctly differentiates between healthy individuals and MM patients when comparing them to control subjects. The cross-validation error rate was 0.09. In contrast to the control and premalignant MGUS comparison, amino acids were crucial in distinguishing the groups. According to Figure 2B, the shift from a healthy to a malignant MM was linked to lipoprotein subfraction variables, specifically HDL-4 (high-density lipoprotein subfraction 4) cholesterol particles, with a VIP score greater than 1.0. On the other hand, during the progression to MM, only methionine, lysine, and leucine amino acids exhibited alterations (Appendix A). Figure 2C presents the ROC curves and boxplots for the top four lipoprotein subfractions variables based on AUC scores for MM and control participants. These included HDL-4 Cholesterol (H4CH) (AUC = 0.99, CI 95% = 0.98–1.0), HDL-4 Phospholipids (H4PL) (AUC = 0.99, CI 95% = 0.98–1.0), HDL-4 Apolipoprotein A-1 (H4A1) (AUC = 0.99, CI 95% = 0.96–1.0), and HDL-4 Apolipoprotein A-2 (H4A2) (AUC = 0.96, CI 95% = 0.92–1.0). The single ROC analysis combining the four significantly altered markers yielded an AUC of 0.97 (95% CI = 0.93–1.00) (Appendix A). Figure 2C also reveals that these lipoprotein subfraction variables were significantly reduced in MM patients compared to healthy controls. Amino acids with a VIP score > 1.0 were incorporated in the pathway analysis in Figure 2D. Impacted pathways included valine, leucine, and isoleucine degradation, purine and pyrimidine metabolism, urea cycle and metabolism of arginine, glutamate, aspartate, and arginine, lipoate metabolism, lysine metabolism, biotin metabolism, folate metabolism, methionine, and cysteine metabolism, as well as nicotinate and nicotinamide metabolism. No significant correlations were found between the clinical data variables (fibrinogen, creatinine, albumin, serum protein, M-protein, CRP, and hemoglobin) and the top four lipoprotein subfractions variables, as indicated in Appendix A.

### 2.4. MGUS vs. MM: Lipoprotein Subfractions Alterations in MGUS Contribute to Symptomatic MM

The PLS–DA plot in Figure 3A shows a clear separation between MGUS and MM groups, with an average cross-validation error rate of ≤0.20. Figure 3B indicates that the significantly altered variables stem from lipoprotein subfractions variables. Figure 3C presents ROC curves and boxplots for the top four lipoprotein subfractions, including HDL Free Cholesterol (HDFC) (AUC = 0.93, CI 95% = 0.86–1.0), Total Apolipoprotein A-1 (TPA1) (AUC = 0.92, CI 95% = 0.84–0.99), HDL Apolipoprotein A-1 (HDA1) (AUC = 0.90, CI 95% = 0.81–0.99), and HDL-3 Cholesterol (H3CH) (AUC = 0.89, CI 95% = 0.80–0.97). Appendix A reveals that all metabolites and lipoprotein subfraction variables with significantly altered *p*-values < 0.001 had AUC scores > 0.86, demonstrating a strong discriminatory ability. The combined ROC analysis on all four significantly modified markers revealed an AUC of 0.90 (95% CI = 0.81–0.99) (Appendix A). Interestingly, HDL-free cholesterol showed the most significant change in the MM disease state, which aligned with the comparison of controls to MM. Glutamine was the only amino acid found to be significantly altered (VIP score > 1.0) in the pathway analysis when comparing MGUS vs. MM (Figure 3D). Glutamine appeared to be connected to several pathways, including purine, pyrimidine, amino sugars, nicotinate, and nicotinamide metabolism, as well as the urea cycle and the metabolism of arginine, glutamate, aspartate, and asparagine. The levels of M-protein and serum proteins showed a strong correlation with HDFC and H3CH. In our study, we have found that these specific subfractions of lipoproteins are markers that are related to the development of MGUS progressing to MM (Appendix A).

## 3. Discussion

In this study, we utilized the ^1^H-NMR analysis to conduct a comprehensive metabolomics analysis on serum samples obtained from healthy individuals, MGUS patients, and MM patients. The goal was to identify metabolites and lipoprotein subfraction variables that might contribute to the development and advancement of MM. Our findings indicated that amino acids are involved in transitioning from healthy controls to asymptomatic MGUS. Conversely, lipoprotein subfractions were the most influential variables for distinguishing between MM and MGUS patient groups.

Our analysis revealed significant changes in amino acid concentrations, such as alanine, lysine, leucine, and formic acid, when comparing healthy controls to MGUS patients. Previous studies have also reported perturbations in amino acid concentrations in MGUS patients [8,21]. However, to our knowledge, this was the first study that has utilized NMR analysis of serum samples to investigate metabolic changes between healthy controls and MGUS patients. In a separate study, Steiner et al. [8] used electrospray ionization liquid chromatography (ESI-LC-MS/MS) and flow-injection analysis mass spectrometry (FIA/MS) to measure peripheral blood plasma samples. They found significant alterations in 36 amino acids and biogenic amines. Unfortunately, the authors did not offer information regarding the particular amino acids that showed significant changes, making it difficult to compare their findings with our own.

Another study by Ludwig et al. [21] analyzed filtered plasma from bone marrow aspirates using ^1^H-NMR spectroscopy. They found that isoleucine was significantly decreased in the bone marrow of MGUS and MM patients, consistent with the findings of our study, which revealed significant changes in the same amino acids. According to Ludwig et al. [21], increased essential amino acid usage by clonal plasma cells within the bone marrow of MGUS patients implies an increase in cellular anabolism. This, in turn, results in greater utilization of branched-chain amino acids (BCAAs) such as leucine, isoleucine, and valine [25]. The observed reduction in BCAA concentrations in the face of enhanced plasma cell proliferation supports this theory. Previous studies have linked BCAAs and their levels to cancer progression, as they are indispensable for cancer cell metabolism, including oxidation and protein synthesis [8,34]. Additionally, amino acid derivatives have been linked to epigenetic regulation of tumorigenesis and metastasis, highlighting the potential significance of BCAAs in the progression to MGUS [35]. Furthermore, the catabolism of BCAAs can promote lipogenesis by producing acetyl-CoA, which is critical, considering the variations we observed when comparing healthy controls to MM patients [35].

As previously stated, the comparison between healthy controls and MM patients revealed a clear distinction between the two groups. The data indicated that lipoproteins were the primary distinguishing factor between healthy individuals and MM patients. It has been recognized that abnormal lipid metabolism is a crucial mechanism in carcinogenesis. Dysregulated lipid metabolism is associated with a poorer prognosis and an increased cancer risk [28,30,36]. Apolipoproteins and cholesterol were the most prominent subfraction variables of lipoproteins that were altered (Figure 2B).

Small clinical studies indicate that the lipid content of lipoproteins is the most prevalent biomarker of MM [30]. In addition, lipoproteins may affect cellular microenvironment processes, such as oxidative stress and inflammation [37]. The significance of lipoproteins in the bloodstream extends beyond their concentration, as their function is equally critical. In particular, HDL is pivotal in several biological processes and pathways, including the redistribution of cholesterol and other lipids in the periphery [30].

According to the literature, apolipoprotein A1 may have an essential role in the progression and development of MM [30,31]. In this study, a metabolomics analysis showed lower levels of total apolipoprotein-A1, HDL-4 Apolipoprotein-A1, and HDL-4 Apolipoprotein-A2 in MM patients compared to healthy controls. Research suggests that high levels of Apolipoprotein-A1 are linked to better overall and progression-free survival [30,31,32,38]. A proteomic analysis by Zhang et al. specifically showed decreased levels of Apolipoprotein-A1 in MM patients compared to controls [38]. Apolipoprotein-A1 is known to have anti-tumor activity and may hinder tumor growth by inhibiting angiogenesis, reducing tumor metastasis and invasion, and regression of tumor size [31,39]. This could be due to its role in cellular cholesterol homeostasis and reverse cholesterol transport [34]. Myeloma cells require cholesterol for growth and proliferation, and previous studies have shown lower cholesterol levels in MM patients. Hungria [40] and Scolozzi et al. [41] reported decreased cholesterol levels in patients with multiple myeloma. In a study by Yavasoglu et al. [34], patients with MM had significantly lower LDL and HDL cholesterol levels than controls.

Hypocholesterolemia in cancer patients may be caused by an enhanced cholesterol uptake by cancer cells [34]. Specifically, low HDL cholesterol levels can result from an impaired HDL metabolic pathway and are also associated with increased deposition [30]. In the bone marrow of patients with multiple myeloma, mature adipocytes are typically disproportionately large, and pre-adipocyte levels are elevated. It is believed that adipocytes support tumor growth and protect malignant cells from chemotherapeutic-induced apoptosis. In this way, lower cholesterol and HDL levels may be linked to the development and progression of multiple myeloma (MM) [30,37,40,41].

The pathway analysis comparing healthy controls and MM patients (Figure 2D) revealed that the metabolism of the urea cycle was modified. This finding was supported by Ludwig et al. [6], who found an increase in anabolism in the microenvironment of MGUS and MM tumors. A distinct pattern emerged when comparing MGUS and MM. The concentrations of lipoproteins, particularly cholesterol, and apolipoprotein exhibited significant alterations. The patterns were comparable in a comparison between healthy controls and MM patients. HDL levels, however, appeared to be more prevalent. As previously mentioned, HDL may be linked or associated with the progression of MM. Interestingly, reduced HDL levels have been linked to the development of an inflammatory microenvironment that affects the function and differentiation of osteoblasts [29]. This may add to the explanation of the elevated bone resorption seen in MM patients [42].

Since changes in HDL levels appeared to be less pronounced in the analysis comparing healthy controls to MGUS patients than in the analysis comparing MGUS to MM, it appears that the most significant shift in HDL metabolism occurred during the progression from MGUS to MM. More specifically, the apolipoproteins were more involved during the malignant progression from MGUS to MM than during the development of MGUS. According to Gonsalves et al. [25], lower levels of complex lipids in the bone marrow plasma of patients with multiple myeloma than in patients with myeloma-associated lymphoid neoplasms indicated an increased utilization of lipids for membrane biosynthesis due to the rapid proliferation of clonal plasma cells. Interestingly, a study found that patients with MGUS and a high BMI were more likely to develop multiple myeloma. This appeared to be related to the fluctuating levels of lipoproteins in MM patients [30]. The disruption of the lipoprotein transport system may have a crucial role in disease development, as indicated by the emerging importance of dyslipidemia as a prognostic factor for disease progression and outcome. The results presented in Figure 1 and Figure 2 showed that metabolites are significantly dysregulated between healthy individuals and those with multiple myeloma (MM) than between healthy individuals and those with monoclonal gammopathy of undetermined significance (MGUS), consistent with previous findings by Ludwig et al. [21]. These authors suggested that alterations to the metabolic phenotype are necessary for disease progression to MM, a notion supported by López-Corral et al. [43], who reported only a few genetic alterations are associated with progression from MGUS to MM.

In our study, MM and MGUS patients correlated with several biomarkers, revealing interesting findings. Specifically, we found a significant positive correlation between creatinine and alanine levels, as well as serum protein and formic acid levels, in the control group compared to the MGUS group. However, no correlation was observed between the control and MM groups. In the MGUS versus MM group, a strong positive correlation with HDL-free cholesterol (HDFC) and a strong negative correlation with HDL-3 cholesterol (H3CH) were observed (Appendix A) [12,25,44,45,46,47,48,49,50]. Hypocholesterolemia, which has been reported in MM patients, was postulated to be a result of increased LDL clearance and cholesterol utilization by myeloma cells. Hyperlipidemic myeloma, a rare variant, may involve the inhibitory role of paraprotein on lipid degradation [12,30,37,40,41,44,45]. Recent studies have highlighted the correlation between protein levels, formic acid, and HDL-free cholesterol in myeloma patients [25,46]. Formic acid, typically associated with methanol poisoning or metabolic disorders, may have toxic effects in MGUS and potentially disrupt lipid metabolism, leading to dyslipidemia-related effects in MM. However, further research is needed to understand its role in myeloma. Alanine and creatinine have gained attention as biomarkers of interest in multiple myeloma. The precise relationship between these markers is not fully understood, but elevated plasma creatinine levels may be linked to decreased renal excretion in multiple myeloma, possibly due to tumor infiltration. Changes in alanine and other branch chain amino acids have been observed in myeloma patients, and lower levels are associated with a poor prognosis for survival [35,51]. Similarly, elevated creatinine levels have been associated with a poorer prognosis in myeloma patients. Additionally, M-protein is associated with an increased risk of kidney dysfunction in MM patients, indicating its potential as a prognostic marker. Monitoring these biomarkers has shown promise in predicting the progression of multiple myeloma, overall survival, and kidney function [25,47,48,49,50,52]. Further research is required to fully comprehend the underlying mechanisms and develop targeted treatments for multiple myeloma. By gaining a deeper understanding of these associations, we may be able to improve the treatment and outcomes for patients with this disease.

It should be noted that this study faced a few shortcomings. First, the sample size was relatively small, which increased the possibility of random chance bias. Increasing the sample size could significantly improve the statistical analysis’ reliability and validity. Second, the research was restricted to NMR metabolomics. As previously stated, using a combination of technologies is highly advantageous for metabolic studies. Implementing mass spectrometry (MS) would permit the detection of metabolites below the detection limit of NMR. Furthermore, incorporating gender stratification in future investigations could be valuable, as previous research has identified notable disparities in the probability of developing MM and overall survival rates between males and females [53]. Serum was obtained from non-fasting individuals, and their lipoprotein profiles differ from those in a fasting state. Nevertheless, the primary distinctions were observed in the HDL fractions, which exhibit minimal alterations between fasting and non-fasting conditions. Additionally, there was an absence of validation using a separate, independent cohort.

In this study, NMR spectroscopy was used for the first time to compare the serum metabolomes of healthy individuals to those of MGUS and multiple myeloma patients. According to Emwas et al. [16], combining different metabolomic analysis techniques produced superior outcomes; hence utilizing various technologies in future research may be beneficial. Future studies can determine the exact significance of apolipoproteins in the development and progression of MGUS and MM by employing a combination of technologies. In addition, analyzing the changes in the metabolome of multiple myeloma (MM) patients in response to different treatments may reveal intriguing mechanisms involved in MM progression and treatment response.

## 4. Materials and Methods

### 4.1. Study Participants

In this cross-sectional study, 20 MGUS and 30 MM newly diagnosed patients, based on the International Myeloma Working Group criteria [1], were recruited at Aalborg University Hospital from the Department of Hematology between March 2015 and September 2017. In addition, MM patients were staged in accordance with the International Staging System criteria (ISS) [54]. Nielsen et al. [33] have previously described the inclusion and exclusion criteria. Briefly, patients were enrolled in the study if they had no history of venous thromboembolism (VTE), prior malignancies, or were receiving anticoagulation therapy (except for acetylsalicylic acid). At the time of diagnosis, clinical and biochemical data (Appendix A) were collected for both MM and MGUS patients and were previously described by Nielsen et al. [33].

For comparison with patient groups, 25 age- and sex-related donors with a mean age of 63 years (range 56–67; 52% males) were recruited from the blood bank at Aalborg University Hospital—blood donors in Denmark are healthy persons without biochemical abnormalities. The study was conducted in agreement with the Declaration of Helsinki and approved by the ethical committee of Northern Jutland (N-20130075). All patients and control subjects provided written informed consent.

### 4.2. Sample Collection and Processing

Blood sampling at the time of diagnosis for MM patients was performed at the outpatient clinic at Aalborg University Hospital by the Department of Clinical Biochemistry. Blood samples were collected in 10 mL clot activator tubes (BD Vacutainer^®^, UK) and centrifuged at room temperature at 2500× *g* for 15 min. The subsequent serum was snap-frozen in liquid nitrogen and stored at −80 °C until analysis.

### 4.3. Biochemical Analysis

Measurements of biochemical parameters, protein concentration, creatinine, C-reactive protein (CRP), albumin, fibrinogen, haemoglobin were performed, as previously described [33].

### 4.4. Nuclear Magnetic Resonance Spectroscopy

Then, ^1^H-NMR analysis was conducted, as previously described by Pedersen et al. [55]. Serum samples (350 µL) were gently mixed with 350 µL of sodium phosphate buffer (0.075 M, pH 7.4, 20% D_2_O in H_2_O, 6 mM NaN3, 4.6 mM 3-(trimethylsilyl)-2,2,3,3-tetradeuteropropanoic acid) (TSP-d4). The prepared samples were then gently mixed and transferred to NMR tubes (5 mm diameter, 40 mm fill height). NMR spectra were collected with a Bruker Avance III 600 MHz spectrometer outfitted with a BBI probe (Bruker Biospin Gmbh, Rheinstetten, Germany). The data acquisition and sample handling were automated using IconNMR on Topspin 3.6.2 and the SampleJet autosampler (Bruker Biospin). Water suppressed ^1^H NMR spectra were recorded at 37 °C using the one-dimensional nuclear Overhauser effect (NOESY) experiment (pulse program “noesygppr1d”) and acquisition parameters from Dona et al. (2014) [56]. The spectra were recorded with 96k data points and 30 ppm spectral width, 32 scans, and with water suppression (25 Hz) during the relaxation delay (4 s) and mixing time (10 ms). After 0 filling to 128k data points and 0.3 Hz line broadening, the free induction decays were Fourier transformed. In accordance with B.I.-Methods (Bruker Biospin Corporation, Billerica, MA, USA), reference samples for temperature calibration, water suppression determination, and external quantitative referencing were routinely recorded and processed in automation (Bruker Biospin). The Bruker Biospin methods B.I.Quant-PS^TM^ 2.0 and B.I.LISA^TM^ [57] automatically calculated quantitative measures of 41 metabolites and 114 lipoprotein subfraction variables (Appendix A). The mean NMR NOESY spectra from control, MGUS and MM is presented in Appendix A. Metabolites with more than 30% measurements below a predefined limit of detection threshold were excluded from the analysis, leaving 24 metabolites for statistical analysis (Appendix A). A comprehensive list of included metabolites/lipoproteins subfraction variables and abbreviations can be found in the Appendix A. To supplement the findings with references and metabolic pathways from existing research, metabolic changes were compared to the Edinburgh Human Metabolic Network [58] and Kyoto Encyclopedia of Genes and Genomes (KEGG) [59,60] databases.

### 4.5. Statistical Analysis

Principle component analysis (PCA) is an unsupervised analysis that transforms a dataset consisting of possibly correlated variables into a new set of uncorrelated variables called principal components. These components are linear combinations of the original variables and are sorted in order of their ability to explain the variance in the data [61]. PCA was used for a dimensionality reduction and visualization of high-dimensional data (Appendix A). To identify metabolic and lipoprotein subfraction variable differences between healthy Controls, MGUS, and MM patients, a partial least squares discriminant analysis (PLS–DA) [62] was performed. PLS–DA is a multivariate analysis technique that combines elements of the Partial Least Squares Regression (PLSR) and Discriminant Analysis (DA) to predict the class membership of observations based on a set of predictor variables. The primary aim of PLS–DA is to build a predictive model that can effectively classify observations into predefined classes based on a set of predictor variables. The data were normalized, and Pareto scaled prior to performing a multivariate analysis. The PLS–DA model was validated using a Cross Validation (CV) model with 10 folds and 1000 permutations. The classification error rates were averaged to determine a single estimate for comparing the balanced error rate of maximum distance and the Mahalanobis distance error rate, which helped signify the optimal number of components to be utilized. The significant metabolites in sample grouping were selected using the Variable Importance in Projection (VIP)-score, where a score of 1.0 indicates significance. Multivariate analyses (PCA and PLS–DA) were conducted with the R-package “mixOmics” and the free software R studio (https://rstudio.com/products/rstudio/download/, accessed on 11 May 2023) [63]. Fold changes (FC) between groups were also calculated for the metabolites, using the formula FC = (Met_MM_/MetS_Con_) or (Met_MGUS_/Met_Con_) or (Met_MM_/MetS_MGUS_).

Prior to univariate analysis, normality testing was performed on the data using a Shapiro–Wilk test. Most data were not normally distributed; therefore, a non-parametric Kruskal Wallis and Mann–Whitney U test was performed to identify metabolites with a post-hoc false discovery rate (FDR) correction test. The median ratio between groups was subsequently subjected to a fold change (FC) analysis. Control vs. MGUS, Control vs. MM, and MGUS vs. MM were compared.

To test the viability of the identified diagnostic metabolites, receiver operating characteristics (ROC) analysis was conducted in the GraphPad Prism version 9.3.0 (GraphPad Software, La Jolla, CA, USA). Receiver operating characteristics (ROC) analysis for combined metabolites was performed using R (v.4.2.1), package pROC, and figures were created using GraphPad Prism 9.5.0. Correlation between lipoproteins and clinical parameters between the different group comparisons was performed using the R packages Hmisc and corrplot [64]. Lastly, visualization of altered metabolite pathways was performed in Cytoscape version 3.9.0 using the Metscape package [51]. All amino acids with a VIP score greater than 1.0 were included in the pathway analysis. The NMR (Appendix A) and clinical data (Appendix A) can be found in the Appendix A presents additional significantly altered metabolites and lipoprotein subfraction variables associated with the progression of myeloma, starting from healthy individuals, to MGUS, to, ultimately, MM. The combined ROC and correlation analyses are in Appendix A, respectively.

## 5. Conclusions

Our study had utilized ^1^H-NMR spectroscopy to reveal a significant rearrangement of amino acids during the development of MGUS. At the same time, lipoproteins, particularly apolipoprotein subfractions, were substantially involved in the progression of MGUS to MM patients. By identifying altered biological pathways not previously detected in plasma or bone marrow aspirates, our findings provided novel insights into the progression of premalignant MGUS to malignant MM. Furthermore, serum metabolic profiling provided valuable information and allowed for the identification of new biomarkers, which can enhance the detection of MGUS and MM, leading to earlier and more effective treatment. Ultimately, our study emphasized the necessity for further scientific exploration into the discovered serum metabolite biomarkers.

## Figures and Tables

**Figure 1 ijms-24-12275-f001:**
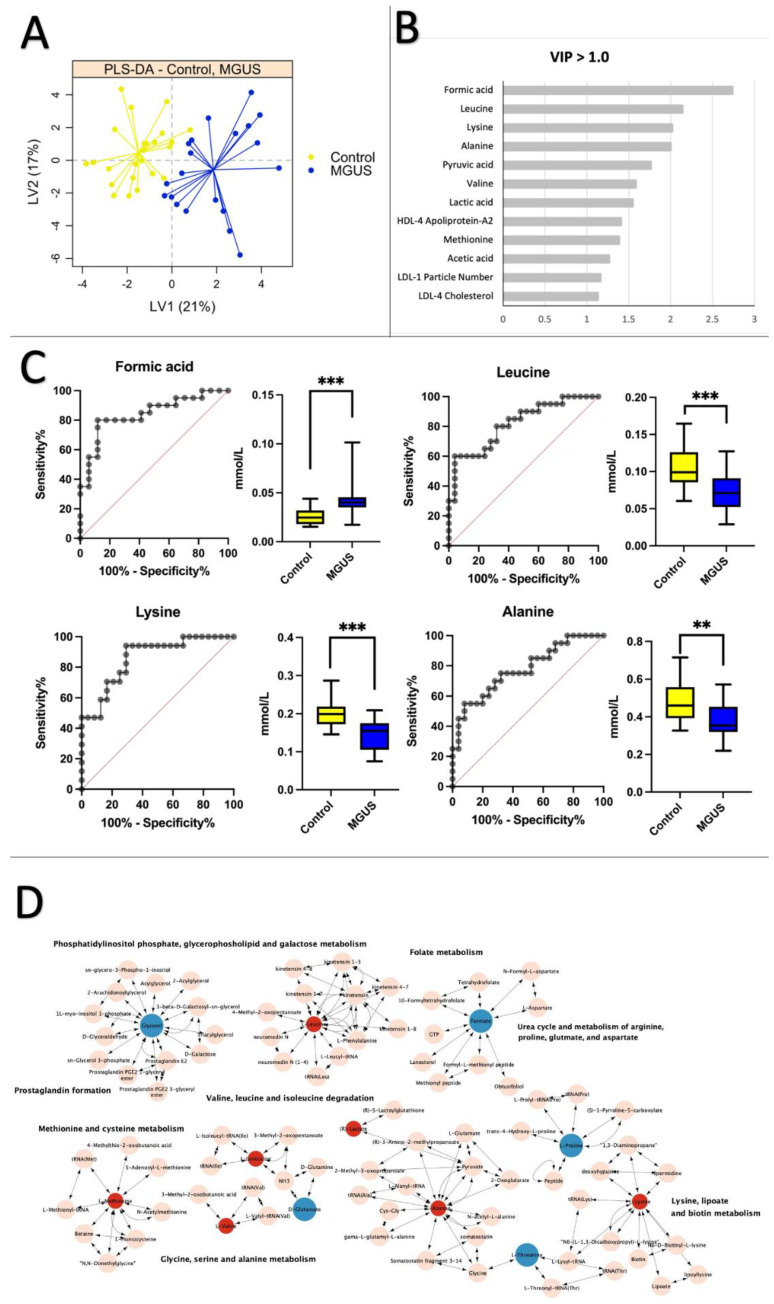
Healthy Control vs. MGUS: Progression of MGUS associated with imbalanced amino acid metabolism. PLS–DA plots, VIP plots, (**A**) partial least squares discriminant analysis (PLS–DA) scores plot comparing healthy control samples (yellow) and MGUS patients (blue) on latent variable 1 (LV1). (**B**) Most significant variables with VIP scores > 1.0. (**C**) OC curves and boxplots for the top four metabolites based on AUC scores when comparing Control vs. MGUS: Lysine (AUC = 0.86, CI 95% = 0.75–0.97), Formic acid (AUC = 0.84, CI 95% = 0.71–0.97), Leucine (AUC = 0.82, CI 95% = 0.69–0.94), Alanine (AUC = 0.78, CI 95% = 0.64–0.91). (**D**) Pathway analysis of significantly altered amino acids (VIP > 1.0) between healthy controls and MGUS patients. Pink nodes represent metabolites involved in the affected pathway but were not investigated in this study. Red nodes denote significantly decreased metabolites, while blue nodes indicate significantly increased metabolites. KEGG IDs were unavailable for all lipoproteins, so only amino acids are included in the pathway analysis. ** *p* < 0.01, *** *p* < 0.001.

**Figure 2 ijms-24-12275-f002:**
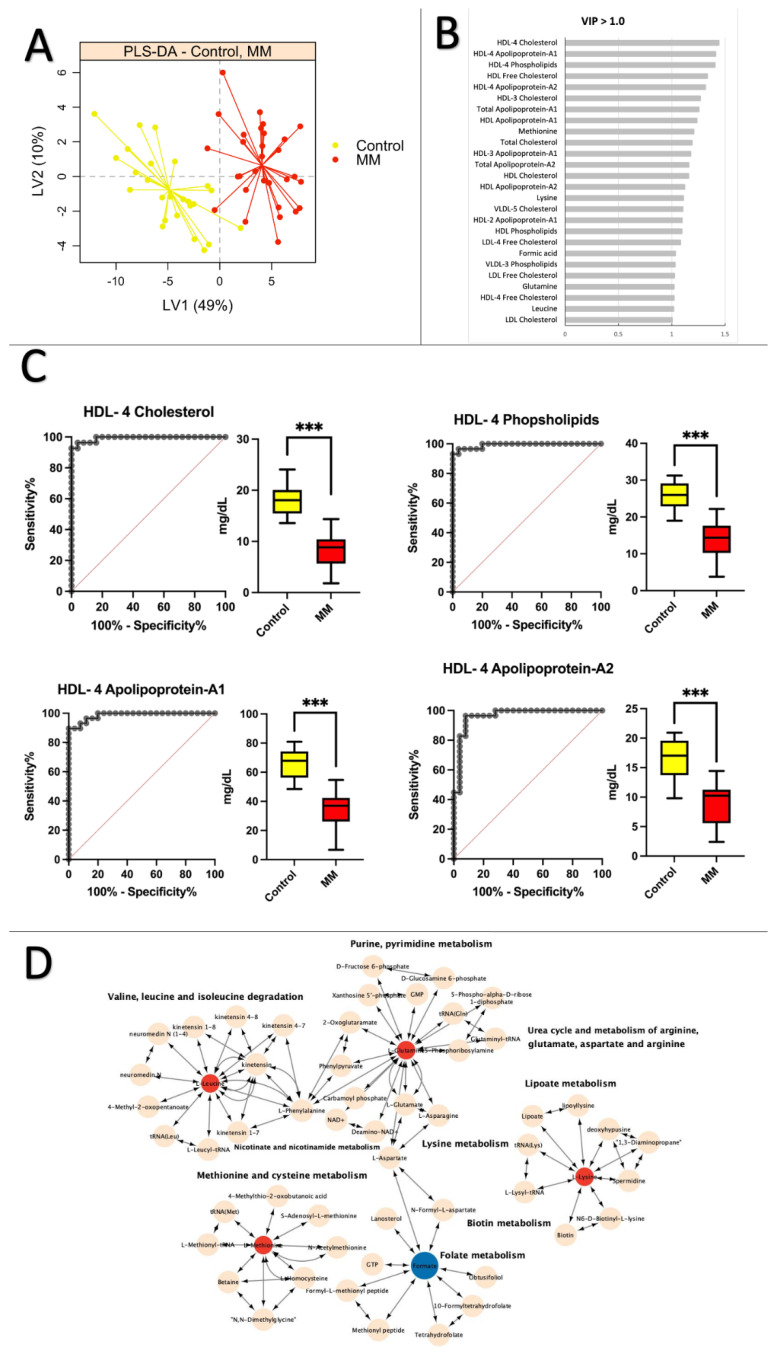
Healthy controls vs. MM: low levels of apolipoprotein and cholesterol were prevalent in MM patients. (**A**) Partial least squares discriminant analysis (PLS–DA) scores plot of healthy control samples (yellow) vs. MM patients (red) on latent variable 1 (LV1). (**B**) Table displays values from the Control vs. MM statistical analysis for variables with a VIP score > 1.3. (**C**) ROC curves and boxplots for the top four variables based on AUC scores when comparing Control vs. MM: HDL-4 Cholesterol (AUC = 0.99, CI 95% = 0.98–1.0), HDL-4 Phospholipids (AUC = 0.99, CI 95% = 0.98–1.0), HDL-4 Apolipoprotein A-1 (AUC = 0.99, CI 95% = 0.96–1.0), HDL-4 Apolipoprotein A-2 (AUC = 0.96, CI 95% = 0.92–1.0). (**D**) Pathway analysis of significantly altered amino acids (VIP > 1.0) between healthy controls and MM patients. Pink nodes represent metabolites involved in the affected pathway but were not investigated in this study. Red nodes denote significantly decreased metabolites, while blue nodes indicate significantly increased metabolites. KEGG IDs were unavailable for all lipoproteins, so only amino acids are included in the pathway analysis. *** *p* < 0.001.

**Figure 3 ijms-24-12275-f003:**
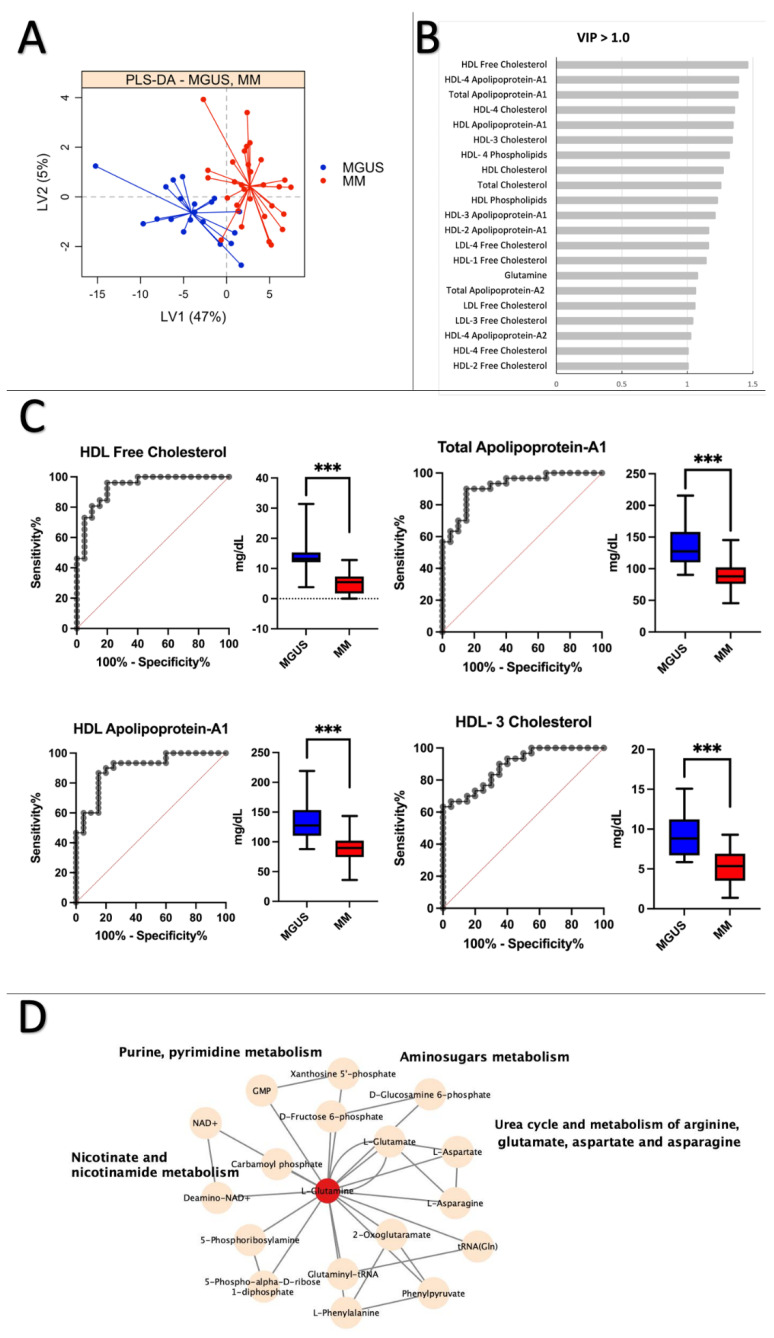
MGUS vs. MM: Lipoprotein subfractions alterations in MGUS contribute to symptomatic MM. (**A**) Partial least squares discriminant analysis (PLS–DA) scores plot of MGUS (blue) vs. patients with MM (red) on latent variables 1 (LV) and 2. (**B**) Most significant variables based on VIP scores > 1.0. (**C**) ROC curves and boxplots of the four highest scoring variables based on AUC scores when comparing Control vs. MM. HDL Free Cholesterol (AUC = 0.93, CI 95% = 0.86–1.0), Total Apolipoprotein A-1 (AUC = 0.92, CI 95% = 0.84–0.99), HDL Apolipoprotein A-1 (AUC = 0.90, CI 95% = 0.81–0.99), HDL-3 Cholesterol (AUC = 0.89, CI 95% = 0.80–0.97). (**D**) Pathway analysis of significantly altered amino acid (VIP > 1.0) between MGUS and MM patients. Pink nodes represent metabolites involved in the affected pathway, which were not investigated in the study. Red nodes represent the significantly decreased metabolites, and blue nodes represent the significantly increased metabolites. *** *p* < 0.001.

**Table 1 ijms-24-12275-t001:** Demographics and clinical information of study populations.

	MGUS (*n* = 20)	MM (*n* = 30)	Between Groups, *p*-Value	Reference Range, Male/Female
**Demographics**				
Age in years (mean ± SD) *	70.35 ± 11	70.7 ± 10	0.996	
Male gender	10 (50%)	14 (47%)		
**Clinical and Biochemical characteristics**				
ISS stage (%)				
I		4 (13%)		
II		16 (53%)		
III		10 (33%)		
Bone changes (%)				
None		8 (27%)		
Halisteresis		0 (0%)		
Localized		3 (10%)		
Spread		19 (63%)		
M-protein, isotype (%)				
IgG	15 (50%)	22 (73%)		
Kappa	8 (53%)	17 (77%)		
Lambda	7 (47%)	5 (23%)		
IgA	4 (20%)	8 (27%)		
Kappa	2 (50%)	6 (75%)		
Lambda	2 (50%)	2 (25%)		
Plasma cells in bone marrow (%)	6.0 ± 2.3	41 ± 19.4	<0.001	
M-protein (g/L)	7.4 ± 6.6	42.9 ± 22.4	<0.001	
κ-Chain, free (mg/L)	128.5 ± 355.3	1179.1 ± 3434.8	0.080	3.3–19.4
λ-Chain, free (mg/L)	26.3 ± 35.0	225.6 ± 652.5	0.014	5.7–26.3
Creatinine (µmol/L)	74.4 ± 26.6	120.2 ± 94.1/87.4 ± 35.6	0.199	60–105/45–90
CRP (mg/L)	7.4 ± 10.9	12.3 ± 25.0	0.812	<8.0
Protein (g/L)	77.2 ± 7.2	107.8 ± 20.0	<0.001	62–78
Albumin (g/L)	36.8 ± 3.1	29.5 ± 4.9	<0.001	34–45
Fibrinogen (μM)	11.4 ± 3.4	10.6 ± 3.9	0.156	5–12
Hemoglobin (M/F) (mmol/L)	8.6 ± 1.3\7.7 ± 0.8	6.4 ± 1.4\5.8 ± 0.7	<0.001	8.3–10.5/7.3–9.5

*** Data is presented as mean ± standard deviation or number (%). Due to gender differences, several parameters are presented as male/female. MGUS = monoclonal gammopathy of undetermined significance; MM = Multiple Myeloma; SD = standard deviation; ISS = international staging system; IgG = immunoglobulin G; IgA = immunoglobulin; κ-Chain, free = kappa-Chain, free; λ-Chain, free = Lamda-Chain, free; CRP = C-reactive protein aminotransferase; ISS = international staging system.

## Data Availability

The mean NMR NOESY spectra from control, MGUS, and MM is presented in Appendix A.

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
