# Peer review of "Serum NMR-Based Metabolomics Profiling Identifies Lipoprotein Subfraction Variables and Amino Acid Reshuffling in Myeloma Development and Progression"

_ijms, 2023, doi:10.3390/ijms241512275_

Round 1
Reviewer 1 Report
This is supposed to be NMR based study (and MS as well). No single NMR spectrum is shown in the manuscript, which I find incredibly bizarre!!
Also:
1- The authors mentioned using 1D NOESY without saying what they learned from the NOESY spectra. This method detects distance restraints, but I find no word about the results or conclusions from this method.
2- The authors mentioned using CPMG, a standard NMR method, to measure relaxation and exchange on the us-ms time regime. Again, no word on what they learned from that. This makes me wonder if any of the authors have expertise in NMR.
Since the paper is about developing NMR methods to study metabolite profiles in MM patients, NMR isn't just a tool here but a primary method that isn't adequately discussed. No need to mention many wrong statements about NMR not requiring sample preparation and limited data processing (the main hurdle in using NMR is data processing).
Reviewer 2 Report
Overall the article is well structured and developed.
The research design is appropriate and the methods are well described.
The patient cohort is not very large, but in the future the statistical model could be validated with an external data set.
The authors collected different clinical parameters for each subject.
Patients with MM showed biochemical abnormalities such as increased levels of protein, creatinine, C-reactive protein (CRP), and decreased albumin, fibrinogen, and hemoglobin.
Why did the authors not conduct a correlation analysis between the alterated clinical data and the discriminated metabolites?
They could obtain additional information for understanding the pathophysiological mechanisms underlying the disease.
Additionally, the authors evaluated the sensitivity and specificity of the discriminant metabolites by ROC curve analysis.
What algorithm was used?
Why not build a single ROC curve by combining the various discriminated metabolites? Combining them could increase the diagnostic potential of a classifier.
Reviewer 3 Report
The research is of interest, the experimental design is solid, the discussion and the conclusions are convincing.
Issues to be addressed:
Line 147: “The mean NMR NOESY spectra from control, MGUS and MM is presented in Supplementary File 2 147 (Figure S1)”. Actually, only some portions of the spectra are presented in supplementary figure S1, full NOESY and CPMG spectra of representative control, MGUS and MM samples should be displayed.
Line 148: “Due to a lack of available measurements, 11 of the metabolites were excluded”. This must be better explained.
Line 324: “This is not consistent with the findings of our study, which revealed significant changes in the same amino acids. Nonetheless, the threonine levels in the MGUS patients from our study were elevated, as indicated by a fold change (FC) of 1.1”. This is very questionable, since data reported in Supplementary file shows that the threonine concentrations of only three MGUS samples out of 20 are non- zero.
Indeed, it is very important to note that the same condition occurs for several metabolites: populations having about 2/3 of zero values are 2-aminobutyric acid (all classes), asparagine (MGUS and MM), ornithine (MGUS and MM), proline (MGUS), sarcosine (all classes), succinic acid (MGUS and MM), acetoacetic acid (controls and MM), glycerol (controls). This fact has two effects: makes the statistical comparison of such classes unreliable, and has an impact on the PLS-DA models since the variance (on which they rely) of these variables is very reduced because of the presence of a big number of data having the same value (zero, in this case). The PLS-DA model should be performed taking those variables out.
Author Response
Response to Reviewer 3 Comments
Open Review
(x) I would not like to sign my review report
( ) I would like to sign my review report
Quality of English Language
( ) I am not qualified to assess the quality of English in this paper
( ) English very difficult to understand/incomprehensible
( ) Extensive editing of English language required
( ) Moderate editing of English language required
( ) Minor editing of English language required
(x) English language fine. No issues detected
|
Yes |
Can be improved |
Must be improved |
Not applicable |
|
|
Does the introduction provide sufficient background and include all relevant references? |
(x) |
( ) |
( ) |
( ) |
|
Are all the cited references relevant to the research? |
(x) |
( ) |
( ) |
( ) |
|
Is the research design appropriate? |
(x) |
( ) |
( ) |
( ) |
|
Are the methods adequately described? |
(x) |
( ) |
( ) |
( ) |
|
Are the results clearly presented? |
(x) |
( ) |
( ) |
( ) |
|
Are the conclusions supported by the results? |
(x) |
( ) |
( ) |
( ) |
Comments and Suggestions for Authors
The research is of interest, the experimental design is solid, the discussion and the conclusions are convincing.
Point 1: Line 147: "The mean NMR NOESY spectra from control, MGUS and MM is presented in Supplementary File 2 147 (Figure S1)". Actually, only some portions of the spectra are presented in supplementary figure S1, full NOESY and CPMG spectra of representative control, MGUS and MM samples should be displayed.
Response 1: In Supplementary File 2 (Figure S1), we have included the full spectra in addition to the highlighted regions.
Point 2: Line 148: "Due to a lack of available measurements, 11 of the metabolites were excluded". This must be better explained.
Response 2: We acknowledge the need for a clearer explanation regarding this matter. The quantifications carried out by B.I.QUANT, as indicated in supplementary Table S1, exhibits numerous zero values. In this context, a zero value in the table signifies that the measured concentration falls below a limit of detection (LOD) that has been empirically determined rather than indicating a concentration of "zero." Based on experiences gained from previous projects, it has been observed that the mentioned LOD is overly cautious, resulting in the exclusion of reliable measurements that lie below the LOD. Consequently, we have made the decision to utilize the raw concentrations provided by B.I.QUANT (refer to supplementary Table S1) as it contains fewer zero values. However, we still employ the LOD as a criterion for selecting which metabolites are included in the statistical analysis. Specifically, a metabolite is excluded if its values fall below the LOD for more than 30% of the samples. To clarify this point, the text has been revised accordingly.
Point 3: Line 324: "This is not consistent with the findings of our study, which revealed significant changes in the same amino acids. Nonetheless, the threonine levels in the MGUS patients from our study were elevated, as indicated by a fold change (FC) of 1.1". This is very questionable, since data reported in Supplementary file shows that the threonine concentrations of only three MGUS samples out of 20 are non- zero.
Indeed, it is very important to note that the same condition occurs for several metabolites: populations having about 2/3 of zero values are 2-aminobutyric acid (all classes), asparagine (MGUS and MM), ornithine (MGUS and MM), proline (MGUS), sarcosine (all classes), succinic acid (MGUS and MM), acetoacetic acid (controls and MM), glycerol (controls). This fact has two effects: makes the statistical comparison of such classes unreliable, and has an impact on the PLS-DA models since the variance (on which they rely) of these variables is very reduced because of the presence of a big number of data having the same value (zero, in this case). The PLS-DA model should be performed taking those variables out.
Response 3: We acknowledge and support the reviewer's feedback. As per response 2, we have addressed the concern by removing the mentioned metabolites from the manuscript and making appropriate revisions to the text. Additionally, we have repeated the PLS-DA model for all three group comparisons utilizing the data from supplementary File 1 (found in the NMR data list). This is illustrated in Figures 1, 2, and 3 of the revised manuscript. We have also revised the text in the discussion section (lines 321-323) that mentions threonine.
Reviewer 4 Report
The paper presented by the authors discussed potential novel insight into the serum metabolic and lipoprotein changes in MM affected patients.
Although a limited number of samples, as the authors wrote, the data could be interesting.
I have some comments about the paper, although already revised as I can see:
Abstract section: Too long, with many details. I think that reporting the FC ratios with p value for each metabolite is not necessary. I suggest to remove it.
Keywords: I think other keywords cold be cited: multivariate analyiis, univariate analysis, pathway analyisis
Introduction: this section is in general well written, with appropriate references.
At line 98: the authors used a nmr-based metabolomics method for their aim, they should wrote it.
section Material and Methods:
2.4 nuclear magnetic resonance: why did the authors substitute cpmpg with noesy experiments? Please justify.
It is not clear which variables were considered for statistical analyisis. NMR data or measures of metabolites and lipoprotein subfraction? What about Nmr spectra processing and bucketing?
2.5 statistical analyisis: please define the used tecniques and add references. Define Multivariate analysis, define pls-da, its aim, and as I suggest in the results section, also define opls-da. The authors should have used opls-da than pls-da for pair wise comparison. It is not clear which values the authors considered for FC ratios calculation: please describe.
All this section is unreferenced, please add appropriate bibliography.
3. Results
3.2,3.3, 3.4: OPLS-DA should be used for pair wise comparison. Moreover, decriptive and predicive parameters of statistical models should be reported.
I also suggest to perform preliminary unsupervised PCA on the whole data set.
In table S1 add the chemical shifts for the metabolites.Moreover, in the same table are indicated four sample groups, why?
A typical 1h NMR spectrum for Controls, MGUS, and MM serum sample, with assignment of principal signals should be added.
minor typos:
line 29: nuclear magnetic resonance should be written as Nuclear Magnetic Resonance.
line 105: the authors should change round with squared brackets.
pay attention to line spacing throughout the manuscript.
Author Response
Response to Reviewer 4 Comments
Open Review
(x) I would not like to sign my review report
( ) I would like to sign my review report
Quality of English Language
(x) I am not qualified to assess the quality of English in this paper
( ) English very difficult to understand/incomprehensible
( ) Extensive editing of English language required
( ) Moderate editing of English language required
( ) Minor editing of English language required
( ) English language fine. No issues detected
|
Yes |
Can be improved |
Must be improved |
Not applicable |
|
|
Does the introduction provide sufficient background and include all relevant references? |
(x) |
( ) |
( ) |
( ) |
|
Are all the cited references relevant to the research? |
(x) |
( ) |
( ) |
( ) |
|
Is the research design appropriate? |
( ) |
( ) |
(x) |
( ) |
|
Are the methods adequately described? |
( ) |
( ) |
(x) |
( ) |
|
Are the results clearly presented? |
( ) |
( ) |
(x) |
( ) |
|
Are the conclusions supported by the results? |
( ) |
(x) |
( ) |
( ) |
Comments and Suggestions for Authors
The paper presented by the authors discussed potential novel insight into the serum metabolic and lipoprotein changes in MM affected patients.
Although a limited number of samples, as the authors wrote, the data could be interesting.
I have some comments about the paper, although already revised as I can see:
Point 1: Abstract section: Too long, with many details. I think that reporting the FC ratios with p value for each metabolite is not necessary. I suggest to remove it.
Response 1: We appreciate your feedback, and we have made the necessary revisions to address this concern. The FC ratios with p-values for each metabolite have been removed from the abstract.
Point 2: Keywords: I think other keywords could be cited: multivariate analysis, univariate analysis, pathway analysis
Response 2: Thank you for your suggestion. We have included the additional keywords as recommended in lines 42-43.
Point 3: Introduction: this section is in general well written, with appropriate references.
At line 98: the authors used a nmr-based metabolomics method for their aim, they should wrote it.
Response 3: Thank you for your valuable suggestion. We have taken your feedback into consideration and have revised the manuscript accordingly. In lines 93, we have now explicitly included the mention of "NMR-based metabolomics" as the method employed for our investigation.
Point 4: section Material and Methods:
2.4 nuclear magnetic resonance: why did the authors substitute cpmpg with noesy experiments? Please justify.
Response 4: Although we actually recorded both CPMG and NOESY spectra in this project, the Bruker IVDr methods (B.I.QUANT and B.I.LISA) both only use NOESY data as input. Therefore, we only describe the NOESY parameters in the manuscript.
B.I.LISA requires NOESY since the signals from lipoproteins are strongly attenuated by the T2 filter in CPMG. Metabolite quantification from NOESY (B.I.QUANT) utilizes spectral deconvolution to handle the broad baseline from macromolecules.
Point 5: It is not clear which variables were considered for statistical analyisis. NMR data or measures of metabolites and lipoprotein subfraction? What about Nmr spectra processing and bucketing?
Response 5: We apologize for the lack of clarity in our previous communication. We would like to clarify that for the statistical analysis, we utilized both metabolites and lipoproteins as part of our study. Metabolites with more than 70% of measurements above the limit of detection are included in statistical analysis (see also response 2 for reviewer 3). Regarding processing, B.I.QUANT uses deconvolution and no binning/integrals for extracting quantitative data. All post-processing details are not provided by Bruker Biospin, but the methods build on reference 38 (Jimenez et al.). This paper also describes the performance of both B.I.LISA and B.I.QUANT in a multilaboratory trial.
Point 6: 2.5 statistical analyisis: please define the used tecniques and add references. Define Multivariate analysis, define pls-da, its aim, and as I suggest in the results section, also define opls-da. The authors should have used opls-da than pls-da for pair wise comparison. It is not clear which values the authors considered for FC ratios calculation: please describe.
Response 6: We appreciate your suggestions. The multivariate analysis techniques used in our study were PCA and PLS-DA, as stated on page 4, under the section Statistical Analysis. We have also included a definition and aim of PLS-DA on page 4, lines 153-155, as requested.
While acknowledging the validity of OPLS-DA as an alternative approach for pairwise NMR comparisons, it is essential to highlight that PLS-DA is a well-established and widely used method for analyzing and classifying NMR spectra to differentiate between sample groups or classes. PLS-DA effectively identifies relevant spectral features to discriminate among the classes of interest by modeling the relationship between NMR spectra (predictor variables) and class labels (class membership), with the goal of maximizing class separation.
It should be noted that OPLS-DA utilizes the same algorithm as PLS-DA but employs a rotation of scores in the scores plot perpendicular/orthogonal to LV1, which enhances score separation. However, in our study, since the class separation achieved by PLS-DA was satisfactory, we did not find it necessary to employ Orthogonal-PLSDA (OPLS-DA) in our results.
Regarding the calculation of FC (fold change) values, they were determined using the raw NMR values for each variable across all subjects in the Control, MGUS, and MM groups. The FC values reported in the article are related to metabolites or lipoprotein subfractions that demonstrated the highest performance based on the results of the ROC curves. The FC formula used is MGUS/Control, MM/Control, or MM/MGUS, depending on the specific comparison being made (refer to page 4, lines 164-165).
Point 7: All this section is unreferenced, please add appropriate bibliography.
Response 7: Thank you for your valuable feedback regarding the unreferenced section. We appreciate your attention to detail and understand the importance of providing appropriate references to support the information presented. In response to your suggestion, we will have added references 42-44 under the Statistical Analysis of the Method section.
Point 8- 3. Results: 3.2,3.3, 3.4: OPLS-DA should be used for pair wise comparison. Moreover, decriptive and predicive parameters of statistical models should be reported.
Response 8: Regarding the choice of OPLS-DA for pairwise NMR comparisons, we acknowledge that it is also a valid option, especially when one wants to emphasize differentiation between two groups. However, in our situation, PLSDA was enough to show a clear separation between groups, therefore, we have not used extra orthogonalization of scores plot on orthogonal-LV1. In addition, it is important to note that PLS-DA is a well-established and widely used method for analyzing and classifying NMR spectra to distinguish between different sample groups or classes, and using orthogonal filtering is not mandatory. In our case, PLS-DA effectively identified spectral features that are relevant for discrimination among the classes. It modeled the relationship between NMR spectra (predictor variables) and class labels and maximized the separation between the classes. OPLSDA and PLSDA use completely the same algorithm, but OPLSDA rotates the scores in the scores plot perpendicular/orthogonal on LV1, and this facilitates a better separation of the scores; however, as the class have been nicely separated by PLSDA, we considered the use of Orthogonal-PLSDA not significant in our results.
LV1 (latent variable 1) influences the descriptive and predictive parameters of PLSDA statistical models. The quality of the LV1 score is assessed based on class clustering, overlap, and interpretability. Combining LV1 interpretation with other statistical metrics like cross-validation, permutation tests, and significance analysis is important for reliable results. LV1 scores are presented in Figures 1a, 2a, and 3a on the x-axis. In our study, the PLS-DA model was validated using 10-fold cross-validation and 1000 permutations. Average classification error rates were compared for the balanced error rate and Mahalanobis distance error rate, determining the optimal number of components for the model.
Point 9: 3. Results
I also suggest to perform preliminary unsupervised PCA on the whole data set.
Response 9: Thank you for this suggestion. The unsupervised PCA using all data has not been included in Supplement File 2, Figure S4.
Point 10: 3. Results
In table S1 add the chemical shifts for the metabolites.Moreover, in the same table are indicated four sample groups, why?
Response 10: We appreciate your feedback. We have addressed the issue by including the chemical shift region utilized for quantifying each metabolite in Supplementary File 2 (Figure S1). We sincerely apologize for any confusion caused by labeling the sample group comparisons in the tables. To provide clarity, Table S3 accurately compares the Healthy control and MGUS groups. Moreover, Table S4 appropriately outlines the comparison between the Healthy control and MGUS groups, while Table S5 specifically focuses on the comparison between MGUS and MM. We acknowledge that there are only three sample group comparisons, and we apologize for any misunderstandings that may have arisen from the previous version.
Point 11: 3. Results. A typical 1h NMR spectrum for Controls, MGUS, and MM serum sample, with assignment of principal signals should be added.
Response 11: Thank you for your valuable suggestion. We appreciate your feedback and have taken it into consideration. In response to your request, we have included the full mean spectra for the Controls, MGUS, and MM serum samples in the supplementary figure. Additionally, we have highlighted the important metabolites in the spectra to provide better clarity and facilitate the assignment of principal signals. We believe that these additions will enhance the comprehensibility of our findings.
minor typos:
Point 12: line 29: nuclear magnetic resonance should be written as Nuclear Magnetic Resonance.
Response 12: Thank you for this suggestion. This has now been revised in Line 28.
Point 13: line 105: the authors should change round with squared brackets.
pay attention to line spacing throughout the manuscript.
Response 13: Thank you for this suggestion. This has now been revised in Line 102.
Round 2
Reviewer 1 Report
The authors stated that they have deposited the raw NMR data and provided a link. Unfortunately, the link does not seem to open the data, indicating that it has not been made public. Consequently, there is no way for me to check or verify the data. I strongly suggest adding some of these NMR spectra in the supplementary file to make them accessible for verification.
Author Response
Response to Reviewer 1 Comments
Comments and Suggestions for Authors
Point 1: The authors stated that they have deposited the raw NMR data and provided a link. Unfortunately, the link does not seem to open the data, indicating that it has not been made public. Consequently, there is no way for me to check or verify the data. I strongly suggest adding some of these NMR spectra in the supplementary file to make them accessible for verification.
Response 1: We appreciate your valuable feedback regarding the accessibility of the raw NMR data for verification purposes. We apologize for any inconvenience caused by the link provided in our previous resubmission. Therefore, we have removed the link and have now included the mean NMR NOESY spectra from control, MGUS, and MM as part of the supplementary material (Supplementary File, Figure S1), allowing for direct access and verification of the data. These spectra provide representative examples that support the findings presented in the paper. Once again, we apologize for any confusion or inconvenience caused by the initial oversight. We greatly appreciate your diligence and attention to detail in reviewing our manuscript, as it has helped us improve the transparency and reliability of our research. Thank you for your valuable input, and we look forward to your continued guidance throughout the review process.
Reviewer 2 Report
I believe that the authors have answered all my questions appropriately.
The conclusions should have been discussed more, and instead have not been changed compared to the previous version. The discovery of new potential altered biological pathways not previously detected could be the subject of possible hypotheses by the authors.
Having said that, I believe that the paper can be accepted in this final version.
Please superscript the 1 of NMR in the discussion section.
Author Response
Response to Reviewer 2 Comments
Comments and Suggestions for Authors
Point 1: I believe that the authors have answered all my questions appropriately. The conclusions should have been discussed more and instead have not been changed compared to the previous version. The discovery of new potential altered biological pathways not previously detected could be the subject of possible hypotheses by the authors. Having said that, I believe that the paper can be accepted in this final version. Please superscript the 1 of NMR in the discussion section.
Response 1: Thank you for your feedback and for taking the time to evaluate our manuscript carefully. We appreciate your thoughtful comments and suggestions. We have carefully considered your points and made the necessary revisions to address your concerns. We are glad to hear that you found our responses to your questions appropriately.
We agree with your suggestion that the discovery of new potentially altered biological pathways could be the subject of possible hypotheses in our discussion. In the revised version, we have included a subsection in the discussion (see page 14, Lines 496-521) where we present several hypotheses that could explain the observed alterations based on the correlation data (Supplementary file 2, Figure S3). These hypotheses are based on the novel pathways identified in our study, and we discuss their potential implications for future research and clinical applications.
Regarding your request to superscript "1" for NMR in the discussion section, we have made the necessary formatting changes as per your suggestion. The reference to NMR is now superscripted as throughout the discussion section (page 12, Lines 402 and 419). We sincerely appreciate your careful evaluation of our manuscript and your recommendation for its acceptance in this final version. We believe that the revisions we have made have significantly improved the paper, addressing your concerns and enhancing its overall quality.
Reviewer 4 Report
The author have improved the manuscript.
Only two minor remaining issues:
1. section 2.5: definitions of PCA, pls-da still need references.
2. I suggest to include a representative 1H -NMR spectrum for controls, MGUS and MM serum sample, not mean spectra.
I recommend publication, after having addressed these points.
Author Response
Point 1. section 2.5: definitions of PCA, pls-da still need references.
Response 1. As requested, the definition of PCA has been included under the method section- statistical analysis. In addition, the requested references 42 for PCA and 43 for PLS-DA have been included in the manuscript.
Point 2. I suggest to include a representative 1H -NMR spectrum for controls, MGUS and MM serum sample, not mean spectra.
Response 2. Thank you for the suggestion. To clarify, the approach to selecting the mean NOESY spectra was to highlight the key spectral features that differentiate the three studied groups from one another and minimize individual biological variations unrelated to the condition. By considering the presence of distinct peaks or regions of interest, we have ensured that the selected spectrum effectively captures the characteristic differences between the groups.
Using mean NMR NOESY spectra provides a concise and informative overview of the spectral distinctions between the groups, facilitating the identification of key features that contribute to the differentiation. This approach is valuable for highlighting the relevant variations and aiding in the interpretation of the NMR data.